# High-Throughput Color Imaging Hg^2+^ Sensing via Amalgamation-Mediated Shape Transition of Concave Cube Au Nanoparticles

**DOI:** 10.3390/nano12111902

**Published:** 2022-06-02

**Authors:** He Zhu, Weizhen Xu, Min Shan, Tao Yang, Qinlu Lin, Kexue Yu, Yanxia Xing, Yang Yu

**Affiliations:** 1National Engineering Laboratory for Rice and By-Products Further Processing, College of Food Science and Engineering, Central South University of Forestry & Technology, Changsha 410004, China; z2013428@sdaeu.edu.cn (H.Z.); 20180100023@csuft.edu.cn (W.X.); 2Technology Center of Gaoqing Black Cattle Product Processing and Quality Improvement, College of Food Science and Engineering, Shandong Agriculture and Engineering University, Jinan 250100, China; z2019036@sdaeu.edu.cn (M.S.); z2017025@sdaeu.edu.cn (Y.Y.); 3Tropical Feed Resources Research and Development Center (TROFREC), Department of Animal Science, Facully of Agriculture, Khon Kaen University, Khon Kaen 40002, Thailand

**Keywords:** mercury, Au nanoparticles, Hg^2+^, sensor

## Abstract

Mercury, as one type of toxic heavy metal, represents a great threat to environmental and biological metabolic systems. Thus, reliable and sensitive quantitative detection of mercury levels is particularly meaningful for environmental protection and human health. We proposed a high-throughput single-particle color imaging strategy under dark-field microscopy (DFM) for mercury ions (Hg^2+^) detection by using individual concave cube Au nanoparticles as optical probes. In the presence of ascorbic acid (AA), Hg^2+^ was reduced to Hg which forms Au–Hg amalgamate with Au nanoparticles, altering their localized surface plasmon resonance (LSPR). Transmission electron microscopy (TEM) images demonstrated that the concave cube Au nanoparticles were approaching to sphere upon increasing the concentration of Hg^2+^. The nanoparticles underwent an obvious color change from red to yellow, green, and finally blue under DFM due to the shape-evolution and LSPR changes. In addition, we demonstrated for the first time that the LSPR of Au–Hg amalgamated below 400 nm. Inspired by the above-mentioned results, single-particle color variations were digitalized by converting the color image into RGB channels to obtain (green+blue)/red intensity ratios [(G+B)/R]. The concentration-dependence change was quantified by statistically analyzing the (G+B)/R ratios of a large number of particles. A linear range from 10 to 2000 nM (R^2^ = 0.972) and a limit of detection (LOD) of 1.857 nM were acquired. Furthermore, many other metal ions, like Cu^2+^, Cr^3+^, etc., did not interfere with Hg^2+^ detection. More importantly, Hg^2+^ content in industrial wastewater samples and in the inner regions of human HepG2 cells was determined, showing great potential for developing a single-particle color imaging sensor in complex biological samples using concave cube Au nanoparticles as optical probes.

## 1. Introduction

Mercury ions have been recognized as one of the most hazardous pollutants to human health; it cannot be decomposed once released into ecosystems, which has received considerable attention [1,2,3]. Moreover, Hg^2+^ can be absorbed by the bodies of animals or plants and then stay inside to contaminate them; it ends up being enriched in the human body through the food chain [4,5]. Once ingested by the human body, Hg^2+^ is not easily excreted and will cause physiological damage, including kidney dysfunction, immune system destruction, and even death [6,7,8]. Therefore, it is of great significance to develop an efficient platform for monitoring Hg^2+^ in biological and environmental systems. Up to now, various techniques have been proposed for Hg^2+^ sensing including coupled plasma mass spectrometry [9,10], atomic absorption spectrometry [11,12], surface-enhanced Raman scattering (SERS) [13], and electrochemical methods [14]. Although the quantification of Hg^2+^ can be achieved by the above sensing methods, these techniques have inevitable limitations, such as time-consuming, inferior reproducibility, poor portability, and requiring professional technicians. Hence, it is highly desired to develop a simple, cost-effective, and highly sensitive approach for Hg^2+^ detection.

Lately, the emerging single-particle detection (SPD) method with dark-field microscopy (DFM) has been widely concerned owing to its high signal-to-noise ratio, good spatiotemporal resolution, and ultralow sample consumption, providing more possibilities for the exploration of special chemical reactions [15]. In addition, single-particle imaging detection provides excellent sensitivity toward analytes and explains the unknown mechanism because each nanoparticle is considered to be the output of a separate signal response and tiny variations occurring on single-particles can be monitored with DFM [16,17,18]. DFM was used to real-time monitor the in situ growth of single Ag@Hg nanoalloys through a direct amalgamation of Ag nanoparticles with elemental mercury. A comprehensive understanding of the growth mechanism of nanoalloys was elucidated: first, Hg atoms were rapidly adsorbed onto Ag nanoparticles; second, the Hg atoms slowly diffuse into Ag nanoparticles; and third, further diffusion of Hg atoms leads to the formation of spherical Ag@Hg nanoalloys [16]. In recent years, many nanoprobes have been successfully used for Hg^2+^ detection and imaging with DFM. For example, Au nanoparticles modified with oligonucleotides have been developed to detect Hg^2+^ levels based on the DFM imaging colorimetric method. In the presence of Hg^2+^, free Au nanoparticles in the solution were bound with the fixed Au nanoparticles due to oligonucleotide hybridization, which results in an obvious color change from green to yellow under DFM [19]. Au nanorods were used as plasmonic transducers for the investigation of Hg^2+^ detection through associated blue shifts of the surface plasmon resonance peak wavelengths (λ_max_) were measured in individual nanorods by DFM [20]. A tetrahedral DNA-directed core-satellite nanostructure is used as the SERS probe to monitor the deteriorating mercury emissions using DFM combined with Raman spectroscopy [21]. The SPD with DFM has high sensitivity in the detection of Hg^2+^. However, the development of high-throughput statistics for Hg^2+^ color imaging detection applications has lagged. To date, the application of SPD with DFM has been proposed for other analytes imaging detection: sulfide, alkaline phosphatase (ALP), and permanganate (MnO_4_^−^) [22,23]. In general, the SPD imaging detection method with DFM provides more possibilities for the exploration of special chemical reactions and brings great opportunities to develop analytical techniques.

Herein, we proposed a high-throughput single-particle color imaging approach for Hg^2+^ detection. As illustrated in Figure 1, in the presence of ascorbic acid (AA), Hg^2+^ is reduced to Hg and forms Au–Hg amalgamate with Au nanoparticles, triggering the morphological change. Under DFM, the concave cube Au nanoparticles experienced a noticeable color variation from red to yellow, green, and finally blue. Encouraged by this result, we quantified the color variation of the single concave cube Au nanoparticles by splitting the image into RGB channels. The (G+B)/R ratio of a large number of individual particles was extracted to quantify Hg^2+^ concentration by Gaussian fitting. Each particle acted as a signal probe and performing Gaussian fitting can effectively avoid the interference of the ensemble averaging effect. A linear dynamic range of 10–2000 nM and the LOD of 1.857 nM were readily achieved. The high sensitivity of this single-particle color imaging was validated by detecting Hg^2+^ in river water, in the inner regions of human HepG2 cells. In summary, a high-throughput single-particle color imaging strategy exhibits promising potential in the field of bio-sensing.

## 2. Experimental Sections

### 2.1. Chemicals

Chloroauric acid (HAuCl_4_·3H_2_O), (3-aminopropyl) triethoxysilane (APTES), and ascorbic acid (AA) were purchased from Sigma-Aldrich (St. Louis, MI, USA). Hexadecyl trimethyl ammonium Bromide (CTAB), disodium hydrogen phosphate (Na_2_HPO_4_), sodium hydrogen phosphate (NaH_2_PO_4_), trisodium citrate (Na_3_C_6_H_5_O_7_), hydrogen peroxide (H_2_O_2_), sodium hydroxide (NaOH), hydrochloric acid (HCl), and nitric acid (HNO_3_) were bought from Beijing Chemical Reagents Company (Beijing, China). Copper sulfate pentahydrate (CuSO_4_·5H_2_O), sodium chloride (NaCl), potassium chloride (KCl), barium chloride (BaCl_2_), chromium chloride hexahydrate (CrCl·6H_2_O), calcium chloride (CaCl_2_), manganese sulfate (MnSO4), lead nitrate (Pb (NO_3_)_2_), zinc chloride (ZnCl_2_), ferric chloride (FeCl_3_), nickel dichloride (NiCl_2_), and mercuric nitrate (Hg (NO_3_)_2_) were purchased from Sinopharm Chemical Reagent Corporation (Shanghai, China). Dulbecco’s Modified Eagle’s medium (DMEM), fetal bovine serum, and 0.25% pancreatin were acquired from the U.S. Gibco Company (Paisley, UK). All glassware used was cleaned in a batch of freshly prepared aqua regia solution (HCl/HNO_3_; 3/1 *v*/*v*), and then rinsed thoroughly with H_2_O before use.

### 2.2. Apparatus

UV-Vis absorption spectra were recorded on a UV-1800 spectrometer (Shimadzu, Kyoto, Japan). Transmission electron microscopy (TEM) images were recorded via a Titan G2 60–300 microscopy (FEI, Thermo Fisher scientific, Hillsboro, OR, USA), and ImageJ software (Version 1.52k, NIH, Bethesda, MD, USA). Ultrapure water with a resistivity of 18.2 MΩ•cm was produced using a Millipore Milli-Q IQ7003 water purification system. A Nikon Ni-U upright microscope equipped with a 100-W halogen tungsten lamp, an oil immersion dark-field condenser (numerical aperture (NA) = 1.20–1.43) and a 40× Plan Fluor objective lens was used for DFM imaging. A DP73 single-chip true-color charge-coupled device (CCD) camera (DP80, Olympus, Tokyo, Japan) was mounted on the left side of the microscope top to capture images. The (G+B)/R ratio of the nanoparticles was processed with ImageJ software.

### 2.3. Preparation of Au Concave Cube Nanoparticles

Typically, 206 μL of HAuCl_4_ solution (1%) and 75 μL NaOH (1 M) were mixed with 10 mL of CTAB in a 50 mL serum bottle. The solution turns pale yellow after being shaken for 30 s. Then, 10 μL of 33% H_2_O_2_ was added to the mixture and the solution color changed from pale yellow to colorless. Next, 300 μL AA (0.1 M) was added, and the solution was kept undisturbed at a 30 °C water bath for 15 min. The solution color varied from colorless to light blue to greenish-blue, indicating the formation of nanoparticles. Subsequently, the excessive unreacted AA and free CTAB were removed by centrifugation twice (12,000 rpm, 5 min) and the nanoparticles were re-dispersed in ultrapure water. Finally, the concave cube Au nanoparticles colloid was stored at 4 °C before use.

### 2.4. Effect of AA, pH, and Reaction Time

The effect of AA, pH, and reaction time was investigated under UV-Vis. For determination of the optimal concentration of AA, briefly, Hg (NO_3_)_2_ solution (with the final concentrations of 200 μM) and 2 mL concave cube Au nanoparticles were added into a 5-mL centrifuge tube. Different concentration of AA solutions (with the final concentration of 10, 20, 30, 40, 50, 60, 70, 80, 90, 100, and 110 μM, respectively) was added to the above- mixed solutions. After gentle shaking, the UV-Vis spectra of the solutions were taken. To determine the optimal pH and reaction time, Hg (NO_3_)_2_ solution and 2 mL of concave cubic Au nanoparticles were added to a 5-mL centrifuge tube. After gentle shaking, the above procedures were repeated by replaced with different pH (5.5, 6.5, 7.5, 8.5, and 9.5) and reaction times, respectively. Then, UV-Vis spectra of the solutions were carried out.

### 2.5. Sensitive and Selective Detection of Hg^2+^ at the Single-Particle Level

For determination of Hg^2+^ sensitivity at the single-particle level under DFM, the concave cubic Au nanoparticles were first absorbed onto the surface of coverslips modified by APTES, followed by soaking in Hg^2+^ solution with a series of various concentrations (e.g., with a final concentration of 0, 10, 50, 200, 500, 1000, 2000, 4000, 20,000, and 40,000 nM, respectively). Subsequently, 80 μM AA was added separately to each above solution and incubated for 10 min. After that, the coverslips were reversely placed on a glass side and the color DFM images were captured with an exposure time of 200 ms. The (G+B)/R ratio by Gaussian fitting was calculated to assess Hg^2+^ content.

To investigate the selectivity toward Hg^2+^, the above procedures were repeated, but Hg^2+^ was replaced with 40,000 nM of other metal ions, including Cu^2+^, Na^+^, K^+^, Ba^2+^, Cr^3+^, Ca^2+^, Mn^2+^, Pb^2+^, and Zn^2+^, and followed by capturing their DFM images. All the experiments were repeated three times.

### 2.6. Hg^2+^ Detection in Industrial Water

To validate the real applications of the assays using concave cubic Au nanoparticles, industrial water was used as the analysis of a real sample, which was collected from laboratory wastewater on the campus of Central South University of Forestry and Technology. Each of the water samples was filtered through a 0.2-µm membrane to remove the potential aggregates and impurities. Subsequently, the coverslips modified by concave cubic Au nanoparticles were soaked in a water sample containing AA (80 µM) for 10 min. Then, the coverslips were reversely placed on a glass side for DFM imaging. For each water sample, three independent experimental tests were performed.

## 3. Results and Discussion

### 3.1. Characterization of Concave Cubic Au Nanoparticles and the Feasibility of the Hg^2+^ Detection

In this work, concave cubic Au nanoparticles were employed as the optical probes to monitor Hg^2+^ content under DFM; thus, the property of the as-obtained concave cubic Au nanoparticles was first characterized. Concave cubic Au nanoparticles have an obvious absorption peak at 730 nm (Figure 1A, the red curve). The solution color is greenish-blue, as shown in the photo (b) (Figure 1A). In the presence of AA and Hg^2+^, the absorption peak of concave cubic Au nanoparticles solution exhibited a blue shift and was located at 370 nm (Figure 1A, the green curve). The blue shift is mainly due to the adsorption of Hg^0^ onto Au nanoparticles surfaces that alter the surrounding dielectric constant [24]. We demonstrated for the first time that the LSPR of Au–Hg amalgamates below 400 nm. Most Au–Hg amalgamates nanoprobes have been developed based on the following three models: the LSPR of Au nanostars, Au nanorods, and Au nanoprisms after amalgamation were shifting to 520 nm, 473 nm, and 620 nm, respectively [25,26,27]. Correspondingly, the color of the solution turned from greenish-blue to brownish yellow, as shown in photo (a), indicating that concave cubic Au nanoparticles have undergone a morphological change (Figure 1A). Under DFM, concave cubic Au nanoparticles exhibited red color. However, after the addition of AA and Hg^2+^, the color of the nanoparticles turns blue (Figure 1B). The concave cubic Au nanoparticles before and after exposure to AA and Hg^2+^ were further characterized with TEM, as shown in Figure 1C,D. TEM images show that the concave cubic Au nanoparticles were uniform in morphology with an average size of 85 ± 1.9 nm, and the Au nanoparticles after the addition of AA and Hg^2+^ displayed good spherical structure with a diameter of 52 ± 1.38 nm. In the presence of AA, it allows reduction of Hg^2+^ to Hg^0^ that subsequently undergoes amalgamation with the concave cubic Au nanoparticles, leading the Au nanoparticles to undergo a morphological change from concave to spherical.

Based on the above results, we aim to develop a high-throughput single-particle color imaging method for Hg^2+^ detection via amalgamation-mediated shape transition of concave cube Au nanoparticles. The feasibility of Hg^2+^ detection was investigated by imaging single nanoparticles under different conditions (Figure 2A). Under DFM, single concave cube Au nanoparticles exhibited bright red. The presence of either AA (final concentration 80 μM) or the Hg^2+^ (final concentration 40 μM), only, did not induce any color change of the concave cube Au nanoparticles. In the presence of 80 μM AA and 40 µM Hg^2+^, the color of the concave cube Au nanoparticles turned blue. This result is also supported by the UV-Vis results, as shown in Figure 2B. The introduction of AA alone (final concentration 80 μM) to concave cube Au nanoparticles solution did not induce any significant change in the UV-Vis absorption spectrum (Figure 2B, b curve). The presence of Hg^2+^ alone (final concentration 200 μM) did not lead to a change in UV-Vis absorbance spectrum (Figure 2B, c curve). Nevertheless, when the concentration of Hg^2+^ is 5 µM in the presence of 80 μM AA, a single UV-Vis absorption peak was blue-shifted (Figure 2B, d curve). With increasing the concentration of Hg^2+^, the absorption spectrum of concave cube Au nanoparticles changed more significantly (Figure 2B, e curve). When the concentration of Hg^2+^ is 100 µM in the presence of 80 μM AA, a single UV-Vis absorption peak at 532 nm was observed (Figure 2B, f curve). If the concentration of Hg^2+^ is 200 µM in the presence of 80 μM AA, the UV-Vis absorbance underwent an increase, accompanied by a blue shift from 532 to 370 nm (Figure 2B, g curve). The results illustrate that we can design a high-throughput single-particle color imaging approach for Hg^2+^ detection via amalgamation-mediated shape transition of concave cube Au nanoparticles under DFM.

### 3.2. Optimization of Experimental Conditions for Hg^2+^ Detection

To obtain the best response performance of concave cube Au nanoparticles for Hg^2+^ detection, key parameters, including pH, the concentration of AA, and reaction time, were systematically optimized, as shown in Figure 3. The Δλ_max_ shift (nm) (the blue-shift distance of the LSPR) was applied to evaluate the response of concave cube Au nanoparticles for Hg^2+^ quantitation under various conditions in this study. Firstly, we explored the suitable concentration of AA in bulk solution. This was achieved by increasing AA concentration from 10 to 110 μM while keeping other parameters constant. As shown in Figure 3A, the UV-Vis absorption peak of concave cube Au nanoparticles gradually blue-shifted with the increase in AA concentration, a plateau was gradually achieved until the amount of AA increased to 80 μM. When the concentration of AA is higher than 80 μM, the Δλ_max_ shift (nm) did not change further, mainly because Hg^2+^ was reduced completely by the AA. Secondly, the solution gradient of pH values from 5.5 to 7.5 (5.5, 6.0, 6.5, 7.0 and 7.5) were tested for Hg^2+^ sensing. The concave cube Au nanoparticles were found to have the greatest response to Hg^2+^ at pH 6.5 (Figure 3B). At pH < 6.5, concave cube Au nanoparticles became unstable, while at higher pH values, the formation of mercuric hydroxides can cause detection effects. Finally, a reaction time was explored (Figure 3C). When AA and Hg^2+^ were introduced in concave cube Au nanoparticles solution, the UV-Vis absorption peak gradually blue-shifted with reaction time prolongs and a steady-state was achieved at ∼10 min. Based on the above results, the optimal analytical conditions for quantitation of Hg^2+^ are 80 μM AA, pH at 6.5, and the reaction time of 10 min.

### 3.3. The Sensitive and Selective Detection of Hg^2+^ at the Single-Particle Level

We designed a sensitive approach for Hg^2+^ detection by high-throughput single-particle color imaging. Although the particles exhibit a noticeable color change from red to yellow, green, and finally blue under DFM, direct determination of color alteration with the naked eye may introduce inevitable artificial error. To digitalize the color variation of nanoparticles, we split the color images into the three channels, red (R), green (G), and blue (B), and the (G+B)/R ratio was extracted to assess color variation. As shown in Figure 4A, concave cube Au nanoparticles exhibited a noticeable red color in the original DFM image and exhibited a noticeable scattering signal in the red and green channel, the spectral response of the color charge-coupled device (CCD) camera in the red and green channel was close to the UV-Vis absorption wavelength of concave cube Au nanoparticles at 730 nm. However, concave cube Au nanoparticles after exposure to Hg^2+^ and AA displayed evident blue color in the original DFM image and the scattering signals in the green and blue channels were enhanced. This was reasonable because the addition of Hg^2+^ and AA led to the LSPR gradually blue-shift of concave cube Au nanoparticles solution, amalgamation led to the shape-evolution of nanoparticles, resulting in the transformation of nanoparticles into spherical nanoparticles located at 370 nm, the results correspond to CCD in the green and blue channel. On this color variation basis, an accurate high-throughput single-particle color imaging strategy can be readily established to monitor the Hg^2+^ by statistically calculating the (G+B)/R ratio with a Gaussian fitting [28].

Single-particle imaging provides higher sensitivity toward analytes compared to traditional bulk measures because each nanoparticle is considered to be the output of a separate signal response and tiny variations occurring on single-particles can be monitored with DFM [15,22]. Under the optimal conditions, we imaged the color variation of concave cube Au nanoparticles with increasing Hg^2+^ concentrations in the range of 0–4000 nM and the (G+B)/R ratio Gaussian fitting was obtained to quantify Hg^2+^ concentration (Over 500 particles for statistics). Figure 5A shows a typical DFM image of the concave cube Au nanoparticles in the range of Hg^2+^ concentration 0–4000 nM, concave cube Au nanoparticles exhibited a uniform red color and changed gradually from red to green and blue color, indicative of the accumulative formation of Au–Hg on the surface of the nanoparticles. The color variation appeared uniform for most of the particles, suggesting that they had similar reactivity toward Hg^2+^ and the single-particle color imaging via amalgamation-mediated shape transition of concave cube Au nanoparticles were indeed successful. A small amount of green color can be distinguished when the Hg^2+^ concentration is higher than 500 nM and all particles turned blue when the Hg^2+^ concentration was above 4000 nM. By determining the (G+B)/R ratio and Gaussian fitting, a dynamic range from 0 to 40,000 μM was obtained in Figure 5B and the corresponding linear equation of (G+B)/R = 0.000718 + 0.381 × C_Hg2+_ was constructed when Hg^2+^ concentration is 10–2000 nM (Figure 5C), with a correlation coefficient (R^2^) of 0.972. The LOD was 1.857 nM (3σ/slope, where σ is the standard deviation of five blank samples). Taken together, the experimental results by using high-throughput single-particle color imaging sensing could be accomplished with the ease of RGB calculation, which is much better than those from colorimetric and fluorescence methods in the linear detection range and LOD (Table 1) [29,30].

To validate the feasibility of our proposed method for practical applications, the selectivity experiment was performed by replacing Hg^2+^ with other metal ions, including Cu^2+^, Na^+^, K^+^, Ba^2+^, Cr^3+^, Ca^2+^, Mn^2+^, Pb^2+^, Zn^2+^, Fe^3+^, and Ni^2+^, followed by capturing their DFM images. As shown in Figure 6A, no obvious color change was observed after adding other metal ions as Hg^2+^ did, confirmed by the DFM images. The (G+B)/R ratio further illustrated that Au formed alloys with Hg more strongly when compared to other metals (Figure 6B). Furthermore, most metal ions cannot be reduced by AA at pH 5.0, but can form chelates with some of them to further reduce their interaction with Au nanoparticles [31]. Therefore, due to specific interaction, our assay demonstrated high selectivity to Hg^2+^, showing the good application in the real sample test.

### 3.4. Real Sample Analysis in Industrial Water

To demonstrate the potential applicability of this strategy for real sample assay, we first investigated the Hg^2+^ content in the industrial wastewater sample. Different Hg^2+^ concentrations (concentrations: 0, 10, 20, and 40 μg/L, respectively) were then added to the river water samples solution containing concave cube Au nanoparticles and AA (80 μM) to obtain recovery of the spiked Hg^2+^. The Hg^2+^ concentrations in the industrial wastewater sample (each with three measurements) were found to be 0.376 μg/L and the sample recovery efficiencies between 102.8 and 107.9% were obtained as depicted in Table 2, demonstrating the reliability of this method for real biological samples.

## 4. Conclusions

In conclusion, we developed a high-throughput single-particle color imaging strategy under DFM for the detection of Hg^2+^ using concave cube Au nanoparticles as optical probes. This strategy is based on the quantification of color variation of the single concave cube Au nanoparticles. In the presence of AA, the Hg^2+^ could be reduced to Hg^0^, which forms Au–Hg amalgamate resulting in a blue shift of the LSPR of nanoparticles. The amalgam reaction was confirmed by TEM and UV-Vis. Under DFM, the concave cube Au nanoparticles undergo an obvious color change from red to yellow, green, and finally blue with the increasing concentration of Hg^2+^. We separated the DFM images into R, G, and B channels and calculated the (G+B)/R ratio to digitize the color change, which is linearly related to Hg^2+^ concentration. A linear range from 10 to 2000 nM (R^2^ = 0.989) and a limit of detection of 1.875 nM were achieved, indicating the high sensitivity of the proposed strategy. In the river water sample assay, recoveries between 102.8 and 107.9% were obtained. Furthermore, we detected the Hg^2+^ level at the inner regions of human HepG2 cells by full use of resolution advantages and the content of Hg^2+^ entering was different attributed to the complex matrix of cells. Therefore, a high-throughput single-particle color imaging strategy provides a powerful platform for sensing and is expected to be extended to other biomolecule assays in clinical samples in the future.

## Data Availability

The data presented in this study are available upon reasonable request from the corresponding author.

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
