# Peer review of "High-Throughput Color Imaging Hg2+ Sensing via Amalgamation-Mediated Shape Transition of Concave Cube Au Nanoparticles"

_nanomaterials, 2022, doi:10.3390/nano12111902_

Round 1

Reviewer 1 Report

The study describes development of analytical procedure for Hg(II) determination by using AuNPs single particle color-imaging strategy under dark-field microscopy (DFM). The study is relatively well designed, structured and performed.

However authors should present novelty of their method. Reduction of Hg(II) and amalgamation on modified or nonmodified AuNPs is well known and already used for Hg sensing even in the absence of ascorbic acid.

Single Gold Nanoparticle-Based Colorimetric Detection of Picomolar Mercury Ion with Dark-Field Microscopy

Xiaojun Liu, Zhangjian Wu,Qingquan Zhang, Wenfeng Zhao, Chenghua Zong, Hongwei Gai*

Plasmonic detection of mercury via amalgam formation on surface-immobilized single Au nanorods

Carola Schopf, Alfonso Martín, Daniela Iacopino a , *

 In introduction literature based on Hg sensing is almost missing. The application of DFM for Hg determination should be discussed instead of other analytes.

The validity of: “In addition, we demonstrated for the first time that the LSPR of Au-Hg amalgamates below 400 nm.” should be confirmed with literature data.

Please check: the corresponding linear equation of (G+B)/R = 300 0.000718 + 0.381×CHg2+ was constructed when glucose concentration?? is 10-2000 nM (Fig- 301 ure 5C), with a correlation coefficient (R2 ) of 0.972.

How detection limit is calculated with so many significant digits?

Linear range could not start from 0 – do not know dependence between 0 and LOD?

Selectivity studies _ Fe(III) and Ni(II) should be included.

Potential applicability for Hg determination in HepG2 cells is not validated.

The whole analytical procedure should be verified with certified reference material.

Added/found method is not suitable in this case. 2.53 nM Hg is too high for river water. The permissible limit for Hg for surface waters is 0.05 µg/L. Please present all values for Hg in µg/L.

Author Response

  1. In introduction literature based on Hg sensing is almost missing. The application of DFM for Hg determination should be discussed instead of other analytes.

Reply: In the second paragraph of the introduction, we have added literatures based on the application of DFM for Hg determination.

  • Single Gold Nanoparticle-Based Colorimetric Detection of Picomolar Mercury Ion with Dark-Field Microscopy. Xiaojun Liu, Zhangjian Wu, Qingquan Zhang, Wenfeng Zhao, Chenghua Zong, Hongwei Gai*

(2) Plasmonic detection of mercury via amalgam formation on surface-immobilized single Au nanorods. Carola Schopf, Alfonso Martín, Daniela Iacopino a*

(3)Tetrahedral DNA-directed core-satellite assembly as SERS sensor for mercury ions at the single-particle level. Feng Ning, Jingjing Shen, Li Chang et al.

  1. The validity of: “In addition, we demonstrated for the first time that the LSPR of Au-Hg amalgamates below 400 nm.” should be confirmed with literature data.

Reply: Most Au-Hg amalgamates nanoprobes have been developed based on the following three models: the LSPR of Au nanostars, Au nanorods, Au nanoprisms after amalgamation were shifting to 520 nm, 473 nm, and 620 nm, respectively [27-29].

  1. Please check: the corresponding linear equation of (G+B)/R = 300 0.000718 + 0.381×CHg2+ was constructed when glucose concentration?? is 10-2000 nM (Fig- 301 ure 5C), with a correlation coefficient (R2 ) of 0.972.

Reply: glucose concentration should be replaced with Hg2+ concentration, which is a word error in the first draft.

  1. How detection limit is calculated with so many significant digits.

Reply: detection limit = 3σ/slope, where σ is the standard deviation of five blank samples.

  1. Linear range could not start from 0 - do not know dependence between 0 and LOD?

Reply: the corresponding linear equation of (G+B)/R = 0.000718 + 0.381×CHg2+ was constructed when Hg2+ concentration is 10-2000 nM.

  1. Selectivity studies Fe(III) and Ni(II) should be included.

Reply: To validate the feasibility of our proposed method for practical applications, the selectivity experiment was performed by replacing Hg2+ with other metal ions, including Cu2+, Na+, K+, Ba2+, Cr3+, Ca2+, Mn2+, Pb2+, Zn2+, Fe3+ and Ni2+. We added Fe3+ and Ni2+ in the lab.

  1. Potential applicability for Hg determination in HepG2 cells is not validated.

Reply: After the consideration of all our authors, we believe that the actual detection application of Hg2+ is of little significance in HepG2 cells. We cancel the detection application of Hg2+ in HepG2 cells, and the detection of Hg2+ in industrial wastewater is more meaningful.

  1. Added/found method is not suitable in this case. 2.53 nM Hg is too high for river water. The permissible limit for Hg for surface waters is 0.05 µg/L. Please present all values for Hg in µg/L.

Reply: The permissible limit for Hg for surface waters in the Chinese National Standard is 1 µg/L instead of 0.05 µg/L. We think it is more meaningful to detect Hg2+ in industrial wastewater, so we detect Hg2+ in industrial wastewater. We present all values for Hg in µg/L.

Reviewer 2 Report

The manuscript-1675542 “High-Throughput color-imaging Hg2+ sensing via amalgamation-mediated shape transition of concave cube Au nanoparticles” by He Zhu , Weizhen Xu , Min Shan , Tao Yang * , Yanxia Xing * , Kexue Yu * , Yang Yudeals presents the results of testing a new highly efficient approach for determining the concentration of Hg2+ ions using amalgamation of gold nanoparticles and determining their amount by of the latest single particle imaging method using dark-field microscopy. The study was well planned and executed.

The results obtained are very interesting.

However, the text of the manuscript requires significant revision. This is especially true for the style of presentation and the quality of the English language. The "Introduction" section is poorly written.

In the case of studying the influence effect of other metal ions on gold nanoparticles, the authors do not provide results on the determination of Hg2+ in their joint presence in the analyzed solution. Will their method also be effective when, in addition to mercury, the medium contains ions of other heavy metals?

Subsection 3.4 requires a more extended presentation. This is especially true for experiments with HepG2 cells.

All comments on the text of the manuscript can be found in the pdf file of the article

Author Response

  1. However, the text of the manuscript requires significant revision. This is especially true for the style of presentation and the quality of the English language. The "Introduction" section is poorly written.

Reply: We have made significant changes to the Introduction section.

  1. In the case of studying the influence effect of other metal ions on gold nanoparticles, the authors do not provide results on the determination of Hg2+in their joint presence in the analyzed solution. Will their method also be effective when, in addition to mercury, the medium contains ions of other heavy metals?

Reply: this method is effective when, in addition to mercury, the medium contains ions of other heavy metals.

  1. 3.Subsection 3.4 requires a more extended presentation. This is especially true for experiments with HepG2 cells.

Reply: after the consideration of all our authors, we believe that the actual detection application of Hg2+ is of little significance in HepG2 cells. We cancel the detection application of Hg2+ in HepG2 cells, and the detection of Hg2+ in industrial wastewater is more meaningful.

Round 2

Reviewer 1 Report

Revision accepted.